# Risk of bias and reporting completeness of randomised controlled trials in burn care: protocol for a systematic review

Amber Young ,[1,2] Barnaby C Reeves ,[3] Hung-Yuan Cheng,[4] Jason Wasiak,[5,6] Duncan Muir,[4] Anna Davies,[7] Jane Blazeby[1,8]

For numbered affiliations see end of article.

**Correspondence to**
Dr Amber Young;
amber.young1@nhs.net

## ABSTRACT

**Introduction** Burn care represents a healthcare and economic burden to patients internationally. Choice of the most clinically effective treatment strategies requires evidence which is best obtained through high-quality randomised controlled trials (RCT). The number of published RCTs of burn care is increasing. However, trial quality and reporting standards are unclear. This study will assess the risk of bias and adequacy of reporting in recent burn care RCTs using tools endorsed by the Cochrane Collaboration.

**Methods and analysis** A systematic literature review will be undertaken, assessing parallel group RCTs evaluating therapeutic interventions for patients with cutaneous burns. Literature searches will use Ovid Medline, Ovid Embase, Web of Science and the Cochrane Library. Separate searches for each database will include medical subject heading and free text terms including 'burn', 'scald', 'thermal injury' and 'RCT'. Two reviewers will independently assess each study for inclusion. Risk of bias (RoB) will be assessed with the revised tool (RoB 2) and reporting completeness with the CONsolidated Standards of Reporting Trials (CONSORT) 2010 guidelines. We will report a narrative synthesis of all studies, including domain specific, and overall risk of bias for the primary outcome of each trial. Inter-rater agreement for RoB 2 will be reported using Fleiss's Kappa. For adherence to the CONSORT guidelines, we will generate a completeness of reporting index for the five domains.

**Ethics and dissemination** No ethics approval is required because published documents will be used. Findings of the study will be disseminated in a peer-reviewed journal and presented at conferences.

**PROSPERO registration number** CRD42018111020.

## INTRODUCTION

In 2009, Chalmers and colleagues estimated clinical research waste to be 85% of global investment.[1] Wastage was attributed to methodological flaws in randomised controlled trials (RCTs). Poor quality RCTs may result in misleading conclusions increasing research waste.[2–4] Numbers of RCTs in burn care research are increasing, as new technologies are regularly introduced.[5–9] However, trial quality and adherence to reporting standards are uncertain.[10–13] Previous studies

### Strengths and limitations of this study

► This systematic review will be the first to assess both the internal validity of burn care randomised controlled trials using the Risk of Bias 2 (RoB 2) tool and adherence to CONsolidated Standards of Reporting Trials 2010 reporting guidelines.
► The inter-rater reliability of RoB 2 will be assessed using six reviewers from different research backgrounds to simulate real-world conditions.
► The systematic review will be limited to 5 years and to articles published in English.
► Assessments of external validity will not be performed.

are limited by their lack of comprehensiveness and methods of quality assessment.[14–16] Quality assessment of RCTs involves evaluating internal and external validity. External validity assesses whether study results can be generalised to other populations.[17] Internal validity, or risk of bias (RoB), is the extent to which the study design is free from bias; a systematic error that leads to a deviation of results from the truth.[18] Transparent reporting is necessary to allow a clear assessment of the trial design and conduct as recommended by the Cochrane Collaboration.[19 20]

Using an objective method to assess RoB and reporting completeness has become increasingly common and considered good practice.[18 21] For assessment of RoB, many tools are available, including overall scores and checklists.[22 23] The Cochrane collaboration endorse a different type of tool based on individual domains.[18] RoB domains are assessed separately, with the overall RoB formed relating to the highest domain score. Accumulating evidence shows that this is the best tool for assessing the RoB of RCTs.[24–26] A revised version of the tool (RoB 2) has been developed which now examines single trial outcomes, includes 'signalling questions' to aid decision-making and a method to allow

an overall RoB judgement.[27] As yet, this tool has not been tested for inter-rater reliability. The Cochrane collaboration recommend 'The CONsolidated Standards of Reporting Trials (CONSORT) statement' to assess adherence to reporting standards.[28]

The aim of this review is to assess the risk of bias and adherence to reporting standards using Cochrane-approved tools. It will also assess inter-rater reliability of the new RoB 2 tool.

## METHODS

This review will meet its aim with these objectives. It will:
► Determine the number of parallel-group, individually randomised trials assessing burn care interventions published over the last 5 years and retrieve included full-text articles.
► Assess the internal validity of the included RCTs using the revised Cochrane-endorsed RoB 2 tool.
► Assess the inter-rater reliability for the new RoB 2 tool.
► Assess the adherence of the included RCTs to CONSORT 2010 reporting guidelines.

### Literature review search strategy

The systematic review will adhere to this prespecified protocol and the Preferred Reporting Items for Systematic Reviews and Meta-Analyses (PRISMA) statement. This protocol will be aligned to the PRISMA-P statement.[29–31] It has been registered with the PROSPERO international prospective register of systematic reviews. We will report any amendments to the protocol that occur while undertaking the study, within the final manuscript.

### Study eligibility
### Types of studies

Included studies must be full-text individually randomised parallel RCTs published in peer-reviewed journals limited to those allocating human subjects to an intervention or control group . We have planned to limit the search to the last 5-year period. We will not attempt to include all burn care RCTs, as would be undertaken in assessment of the effect of an intervention. Instead, we will investigate whether recent publications of burn care RCTs adhere to the CONSORT statement when reporting their findings and whether the trials we include are at low risk of bias according to the revised Cochrane RoB 2. We will exclude RCT protocols, conference proceedings, abstracts, non-English language publications, interim analysis reports and studies not involving human subjects. Health economic evaluation reports of clinical trials will be considered if they contain enough information for assessing risk of bias (eg, clearly described methods for the trial conduct). We will also exclude trials that compare treatments within subjects as there is not yet a RoB 2 tool designed for assessing such trials. Also, in burn care research, these are typically not cross-over trials as generally defined.[32 33] Instead, they commonly use two wounds or two parts of the same wound. It is uncertain what the dependence between these might be or how treatments might 'contaminate' one another.

### Types of participants

We will include studies evaluating two or more interventions in patients of any age with cutaneous burns. Studies where the population consists of patients with combined burn and mechanical injuries will be excluded, as the data relating to burn patients alone are likely to be difficult to disaggregate.

### Type of interventions

Interventions to treat cutaneous burns of any aetiology.

### Types of outcome

Clinical or patient-reported outcomes. Laboratory studies will be excluded.

### Identification of studies

A predefined search strategy previously designed by the authors in conjunction with experienced systematic reviewers to identify RCTs in the field of burn care will be used. Electronic searches of Ovid Medline, Ovid Embase, Web of Science and The Cochrane Library will be searched using medical subject heading and free-text terms including 'burn', 'scald' 'thermal injury' and 'RCT'. To limit the search to RCTs, we will use terms derived from published RCT search strategies on Medline and the BMJ best practice guideline.[34 35] The thesaurus vocabulary of each database will be used to adapt the search terms. The search strategy for Ovid Medline is included in a previous publication and in Table 1.[36]

### Study selection process

Before screening abstracts and full-text papers, authors undertaking study selection, RoB 2 assessment and data extraction, will undergo training to ensure a comparable understanding of the purpose of the review and the eligibility criteria. The reference management software EndNote (Endnote X8 Clarivate Analytics) will be used

---

| **Box 1   Ovid Medline search strategy** |
|---|
| Ovid Medline search strategy |
| 1.  Burns/(MESH) exp |
| 2.  Burn*.tw |
| 3.  Scald*.tw |
| 4.  Thermal* adj injur*.mp |
| 5.  or/1–4 |
| 6.  Heartburn.mp |
| 7.  Burnout.mp |
| 8.  (Burn* adj out).mp |
| 9.  Burning.mp |
| 10.  Burnetii.mp |
| 11.  Burnish*.mp |
| 12.  Burnet*.mp |
| 13.  6 OR 7 OR 8 OR 9 OR 10 OR 11 OR 12 |
| 14.  5 NOT 13 |
| 15.  Limit to RCT, clinical trial, English language, human, last five years |

## Box 2 Reasons for full-text exclusion

- ► Duplicate.
- ► Not published within time period.
- ► Population not consisting of burn patients alone.
- ► Not a parallel group RCT.
- ► Not in English.
- ► Non-human/animal study.
- ► No full text available.
- ► Laboratory-based study.
- ► Volunteer study.
- ► Oesophageal burns only.
- ► Ocular burns only.
- ► Anaesthetic/sedation technique only (pain management included).
- ► Smoke inhalation injuries without an associated cutaneous burn.
- ► Diagnostic test trial.
- ► Protocol only.

to compile all titles derived from the initial searches, with duplicates removed, for the review authors to screen titles and abstracts against the eligibility criteria. Screening of titles and abstracts will be completed by one reviewer (AY). Of these, 20% will be checked by another reviewer independently (DM). Any studies appearing to meet the inclusion criteria based on the abstract will be retrieved as full-text articles. Two reviewers will then read the full-text articles in their entirety to assess for eligibility, with decisions on inclusion and exclusion recorded (AY,DM). Reasons for exclusion will be ordered hierarchically from most to least important (box 2) and applied to each full-text paper. The most important reason for exclusion met by a paper will be recorded as the reason for exclusion. Any disagreements will be discussed with senior researchers (JB, BR).

### Data extraction and analysis

All studies will be assessed by two reviewers independently and then in duplicate (AY, DM), with disagreements resolved by discussion until consensus is reached or by a senior reviewer (BR). Data will be extracted into a standardised data extraction Microsoft Excel spreadsheet which will be specifically designed for this study. It will be adapted from the RoB 2 tool to include details for the CONSORT checklist by one of the coauthors (H-YC). Before starting the assessment, reviewers will undergo training with the spreadsheet. The reviewers will then independently pilot test each tool on five RCTs. If there are significant differences in the application of the tools in the pilot round, additional testing will be undertaken.[37]

Data extracted will include study details and research design. Study details will include author, publication year, number of sites and number of participants recruited per trial, design (full RCT, pilot study) and intervention tested. Risk of bias judgement will be reported for all trial primary outcomes at domain and overall level. Completeness of reporting will be reported for all RCTs.

### Assessment of risk of bias

Included studies will be objectively assessed for internal validity using RoB 2.[38] The tool has five domains to assess bias arising from randomisation, deviations from intended interventions, missing outcome data, outcome measurement and in selection of the reported result. The assessment is specific to a single trial outcome. Categories of the overall risk of bias for the study outcome are low (risk of bias is low for all domains), some concerns (some concerns in at least one domain) and high (high risk of bias for at least one domain or some concerns for multiple domains).

### Choice of outcome

Risk of bias will be assessed for the treatment effect for the primary outcome in each included trial. If the primary outcome is not reported explicitly, we will use the following decision rule to select the treatment effect to report: we will assess the treatment effect for the outcome used to calculate sample size and, if this too is not reported, assess the treatment effect for the outcome named in the title, then the first reported outcome in the results.[37]

### Assessment of the inter-rater agreement of the RoB 2 tool

As RoB 2 is new, measuring agreement between reviewers will help assess whether the new guidance with signalling questions has improved the reliability of the tool compared with the previous version.[39–43] To evaluate the inter-rater agreement, six independent reviewers will assess the same 30 studies allocated in a balanced incomplete blocks design (please see Appendix A), ensuring that each study is rated 3 times, each assessor rates 15 studies and all pairs of assessors rate 6 studies. Inter-rater agreement will be measured for each domain of bias and for the overall RoB judgement by calculating Fleiss's Kappa scores.[44 45] We will categorise agreement as poor (0.00), slight (0.01–0.20), fair (0.21–0.40), moderate (0.41–0.60), substantial (0.61–0.80) or almost perfect (0.81–1.00).[26]

## ASSESSMENT OF REPORTING COMPLETENESS

Completeness of reporting for each included RCT will be assessed using the latest revision of the CONSORT statement checklist.[46] For assessing completeness of reporting, we will calculate a reporting index defined as the percentage of items reported in each of five domains (title/abstract, introduction, methods, results and discussion), as has been done previously.[47]

### Patient and public involvement

There was no patient or public involvement in designing the study or writing up the study protocol.

**Author affiliations**
[1]Bristol Centre for Surgical Research, Population Health Sciences, Bristol Medical School, University of Bristol, Bristol, UK

[2]Paediatric Anaesthesia, University Hospitals Bristol NHS Foundation Trust, Bristol, UK
[3]Bristol Trials Centre (BRI-Hub), Bristol Medical School, University of Bristol, Bristol, UK
[4]Population Health Sciences, Bristol Medical School, University of Bristol, Bristol, UK
[5]Olivia Newton John Cancer Wellness & Research Centre, Department of Radiation Oncology, Austin Health, Heidelberg, Victoria, Australia
[6]Austin Health Clinical School of Nursing, Latrobe University, Heidelberg, Victoria, Australia
[7]Centre for Academic Child Health, University of Bristol, Bristol, UK
[8]NIHR Biomedical Research Centre, University of Bristol and University hospitals Bristol NHS Foundation Trust, Bristol, UK

**Contributors** AY wrote the paper and conceived the project with the support of JB and BR. H-YC devised the interactive study-specific data extraction spreadsheet. DM will contribute to data extraction and assisted in writing this paper. JB, BR, JW, AD and H-YC edited and critically revised the article. All authors have read and approved the manuscript. AY is the guarantor of the review.

**Funding** This work was supported by the National Institute for Health Research Doctoral Research Fellowship DRF-2016-09-031. The views expressed are those of the author(s) and not necessarily those of the National Health Service (NHS), the National Institute for Health and Research (NIHR) or the Department of Health and Social Care. JB is partly funded by the Bristol Biomedical Research Centre at the University Hospitals Bristol NHS Foundation Trust and the University of Bristol; Grant no. BRC-1215-2001. JB is an NIHR Senior Investigator.

**Disclaimer** The views expressed are those of the author(s) and not necessarily those of the NHS, the NIHR or the Department of Health and Social Care.

**Competing interests** None declared.

**Patient consent for publication** Not required.

**Provenance and peer review** Not commissioned; externally peer reviewed.

**ORCID iDs**
Amber Young http://orcid.org/0000-0001-7205-492X
Barnaby C Reeves http://orcid.org/0000-0002-5101-9487

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
