## [Reviewer comments · BMJ Open]

ARTICLE DETAILS

TITLE (PROVISIONAL)	Risk of bias and reporting completeness of randomised controlled trials in burn care: Protocol for a systematic review.
AUTHORS	Young, Amber; Reeves, Barnaby; Cheng, Hung-Yuan; Wasiak, Jason; Muir, Duncan; Davies, Anna; Blazeby, Jane

VERSION 1 – REVIEW

REVIEWER	CHAUVIN Anthony Assistance Publique hôpitaux de Paris- France
REVIEW RETURNED	11-Sep-2019

GENERAL COMMENTS	The protocol is clearly writing and interesting. Authors adhered to the PRISMA checklist and registered the study in PROSPERO. But, i have some remarks. 1. Authors must clearly explain and justify in the section method why they limit the literature review to to last five years (Table 1, page 8). What mean the 5 last years?2. Authors clearly reported that they will evaluate the completeness of the reporting. But it could be interesting to evaluate if select manuscripts reported a switch in outcomes. Did authors discuss that? Switch outcome clearly influence the quality of study and results.3. In the data extraction and analysis section (P10), Authors must specified that the data extraction form will be standardized.
---

REVIEWER	Mohammad Ali Mansournia Tehran University of Medical Sciences, Iran
REVIEW RETURNED	19-Sep-2019

GENERAL COMMENTS	This is a good methodological study. I only suggest that the authors also assess "sparse data bias" which can threaten both randomized trials and systematic reviews of randomized trials; please see the following reference: https://www.bmj.com/content/352/bmj.i1981
---

VERSION 1 – AUTHOR RESPONSE

Reviewer: 1

Reviewer Name: CHAUVIN Anthony

Institution and Country: Assistance Publique hôpitaux de Paris- France

Please state any competing interests or state 'None declared': None declared

Please leave your comments for the authors below

The protocol is clearly writing and interesting. Authors adhered to the PRISMA checklist and registered the study in PROSPERO.
But, i have some remarks.

We thank the reviewer for their helpful and interesting comments.

1. Authors must clearly explain and justify in the section method why they limit the literature review to to last five years (Table 1, page 8). What mean the 5 last years?

Thank you. This is an important point.

We have limited the search to the last five year period because we believe it is important to limit the assessment of the quality of methodology and outcome reporting to contemporary publications, so that the findings are relevant to researchers now. We also believe that it is very unlikely that papers published prior to this period are lower in quality than those in our review. In addition, in undertaking this work, we will not attempt to be comprehensive in exploring all burn care RCTs, as we are not aiming to assess the effect of an intervention, but rather aiming to investigate whether recent burn care RCTs adhere to the CONSORT Statement when reporting their findings and whether recent trials are at risk of bias according to the revised Cochrane risk-of-bias assessment tool (ROB 2). We have altered the text in the methods as below:

Page 4: "We have planned to limit the search to the last five year period. We will not attempt to include all burn care RCTs, as would be undertaken in assessment of the effect of an intervention. Instead, we will investigate whether recent publications of burn care RCTs adhere to the CONSORT Statement when reporting their findings and whether the trials we include are at low risk of bias according to the revised Cochrane risk-of-bias assessment tool (ROB 2)."

2. Authors clearly reported that they will evaluate the completeness of the reporting. But it could be interesting to evaluate if select manuscripts reported a switch in outcomes. Did authors discuss that? Switch outcome clearly influence the quality of study and results.

We thank the reviewer for this interesting comment. Switching outcomes is embedded in both CONSORT (Altman 2017) and ROB 2 tools. ROB 2 explicitly requires assessors to describe the documents about a trial that have been reviewed (e.g. protocol, statistical analysis plan). Both tools have relevant questions for detecting this issue. We will report on this as part of our assessment of the included studies using both tools.

3. In the data extraction and analysis section (P10), Authors must specify that the data extraction form will be standardized.

We agree with the reviewer and will add this to the Methods section as per the below:

Page 7: "Data will be extracted into a standardised data extraction Microsoft Excel spreadsheet which will be specifically designed for this study."

Reviewer: 2

Reviewer Name: Mohammad Ali Mansournia

Institution and Country: Tehran University of Medical Sciences, Iran

Please state any competing interests or state 'None declared': None declared

Please leave your comments for the authors below

This is a good methodological study. I only suggest that the authors also assess "sparse data bias" which can threaten both randomized trials and systematic reviews of randomized trials; please see the following reference:

<https://www.bmj.com/content/352/bmj.i1981>

We thank the reviewer for their important comment and the interesting link to the paper. This issue ("The presence of only a small number of events of interest for each clinical variable that is being studied, in particular if all (or nearly all) of the events fall into one study group and none (or nearly none) fall into the other"[1]). However, we are not actually analysing the results from the studies, merely the methodological quality. This issue will therefore not form a key part of the results. However, it would be a useful and interesting issue to analyse in the future and we thank the reviewer again for pointing this out.